# Ecological Responses to Climate Change and Human Activities in the Arid and Semi-Arid Regions of Xinjiang in China

Yanqing Zhou [1,2,3,4], Yaoming Li [2,3] , Wei Li [1,2,3,4], Feng Li [2,3,4] and Qinchuan Xin [2,3,5,*]

1. College of Ecology and Environment, Xinjiang University, Urumqi 830046, China
2. State Key Laboratory of Desert and Oasis Ecology, Xinjiang Institute of Ecology and Geography, Chinese Academy of Sciences, Urumqi 830011, China
3. Research Center for Ecology and Environment of Central Asia, Chinese Academy of Sciences, Urumqi 830011, China
4. University of Chinese Academy of Sciences, Beijing 100049, China
5. Guangdong Key Laboratory for Urbanization and Geo-Simulation, Sun Yat-sen University, Guangzhou 510275, China
* Correspondence: xinqinchuan@ms.xjb.ac.cn

**Abstract:** Understanding the impacts and extent of both climate change and human activities on ecosystems is crucial to sustainable development. With low anti-interference ability, arid and semi-arid ecosystems are particularly sensitive to disturbances from both climate change and human activities. We investigated how and to what extent climate variation and human activities influenced major indicators that are related to ecosystem functions and conditions in the past decades in Xinjiang, a typical arid and semi-arid region in China. We analyzed the changing trends of evapotranspiration (ET), gross primary productivity (GPP) and leaf area index (LAI) derived from the Moderate-Resolution Imaging Spectroradiometer (MODIS) satellite product and the Breathing Earth System Simulator (BESS) model in Xinjiang for different climate zones. We separated and quantified the contributions of climate forcing and human activities on the trends of the studied ecosystem indicators using the residual analysis method for different climate zones in Xinjiang. The results show that GPP and LAI increased and ET decreased from 2001 to 2015 in Xinjiang. Factors that dominate the changes in ecosystem indicators vary considerably across different climate zones. Precipitation plays a positive role in impacting vegetation indicators in arid and hyper-arid zones and temperature has a negative correlation with both GPP and LAI in hyper-arid zones in Xinjiang. Results based on residual analysis indicate that human activities could account for over 72% of variation in the changes in each ecosystem indicator. Human activities have large impacts on each vegetation indicator change in hyper-arid and arid zones and their relative contribution has a mean value of 79%. This study quantifies the roles of climate forcing and human activities in the changes in ecosystem indicators across different climate zones, suggesting that human activities largely influence ecosystem processes in the arid and semi-arid regions of Xinjiang in China.

**Keywords:** ecosystem indicators; climate change; human activities; climate zones

## 1. Introduction

According to the Sixth Assessment Report of the Intergovernmental Panel on Climate Change [1], the global climate has warmed significantly since the industrial revolution, and global precipitation will increase but with significant regional and seasonal variations. Global air temperatures increased by at least 0.2 °C per decade over the past 30 years [2]. Models and observation results show that climate change significantly affects vegetation dynamics, ecosystem productivities, and water resource distributions [3,4]. Drylands cover about 41% of Earth's land surface and support over 38% of the population [3,4]. Dryland ecosystems are considered fragile and sensitive to climate change [4]. Aridity increases attributed to climate change would lead to a serious decline in ecological security. In

addition, there are negative environmental consequences such as soil moisture limitation and interruption of biogeochemical cycles [5]. These changes are inconsistent with the observed greening trend on many drylands [6]. Xinjiang is a large inland province in northwest China and has large arid and semi-arid areas in inland Asia. In recent years, climate warming and precipitation reduction in Xinjiang in northern China had large impacts on the ecosystem [7,8]. Climate warming could accelerate evapotranspiration and enhance soil moisture loss [9]. The reduced precipitation is expected to affect ecosystem function and stability [10]. Climate change appears to be detrimental to the growth of vegetation in Xinjiang. However, the vegetation was increasing in Xinjiang from 1989 to 2011 [11]. Research on how terrestrial ecosystems respond to climate variation has gained attention from scholars [9,12–14]. Land cover and land use changes (e.g., farming and grazing), as well as society changes (e.g., rising population pressure), also have significant impacts on ecosystems [15]. It is important to improve our understanding of the roles of both climate change and human activities in affecting ecosystems in Xinjiang.

Vegetation is an essential component of terrestrial ecosystems as it affects water, carbon, and energy exchanges between terrestrial ecosystems and the atmosphere [16]. As a result, the vegetation response to climate variation has been employed as a general indicator to assess the terrestrial ecosystem conditions. Many satellite-based studies used the normalized difference vegetation index (NDVI) derived from remote sensing data to analyze the spatiotemporal changes in vegetation cover [9,13,14]. Using NDVI is suitable for exuberant vegetation, and soil background could influence the NDVI [17]. Considering the low vegetation cover in Xinjiang, leaf area index (LAI) is used to indicate the vegetation change. LAI is defined as total plant leaf area per unit of land area [18,19], which is an important structural parameter in terrestrial ecosystems and is directly related to the growth conditions of vegetation [17,20]. Terrestrial ecosystems have massive exchanges with the environment for energy and materials such as water and carbon [21–23]. Changes in vegetation are bound to affect the carbon and water cycle process on the land surface [16]. Gross primary production (GPP) is the total carbon sequestration, which is an important component of carbon cycling, and it reflects the ability of vegetation to absorb carbon from the atmosphere. Terrestrial evapotranspiration (ET) is a crucial component in the hydrological cycle and energy balance, and it reflects the ability of a vegetated land surface to release water into the atmosphere. Exploring the changes and driving mechanisms of ecosystem indicators has become a focus of global change and is essential for evaluating the evolution of terrestrial ecosystems.

Many studies have investigated the relationship between vegetation dynamics and climate factors to explore the underlying mechanisms for ecosystem indicator change [9,16,24]. Temperature and precipitation are generally considered two key climate factors that control vegetation growth and photosynthetic activities [9,24,25]. The rate of photosynthesis and respiration in vegetation is affected by temperature. Vegetation growth is sensitive to precipitation variations in arid and semi-arid areas. Previous studies have shown that precipitation is the most important factor that controlled vegetation growth in the central and northernmost regions of Xinjiang from 2000 to 2010 [11]. Vegetation growth has a closer correlation with precipitation than temperature in Xinjiang [24,26], and climate warming might prevent vegetation growth in Xinjiang [25]. The relationship between climate and ecosystem indicators could vary based on the extent of human activities and across different climate zones [27,28]. However, the factors dominating ecosystem indicator changes in Xinjiang across different climate zones are still unknown. Both climate change and human activities impact ecosystems, which may affect the spatial differences in ecosystem patterns. Although many studies examined the correlation between ecosystem indicators and climate factors, there is a need to examine how the relationship between ecosystem indicators and climate factors varies across climate zones. In addition, human activities could likely impact vegetation growth in arid and semi-arid regions. There is a need to enrich our understanding of the impact of human activities on ecosystems for sustainable development. In essence, it is meaningful to improve our understanding of the role of

climate change and human activities in vegetation changes for regional developments. Quantitatively assessing the impacts of climate change and human activities on ecosystems is still a challenge that involves mathematical and statistical methods, including PCA (principal component analysis) and correlation analysis [29]. Uncertainty exists in both the processes and the driving factors of the ecosystem change. A single-scale analysis of driving factors may misestimate the actual impact of influencing factors. Residual analysis can avoid the disadvantages of single-scale analysis. This method has been used in studying different types of ecosystems [9,12,30–33] and across different regions, such as the Loess Plateau [9], Central Asia [13], the Belt and Road Initiative Region [30] and Africa [34].

Both temperature and precipitation in Xinjiang have increased in recent years [8,11], and the rate of increase in temperature is much higher than the global level [7]. The local ecosystem in arid and semi-arid areas is fragile and particularly sensitive to climate change [8,35–38]. In addition, over the past half-century, Xinjiang witnessed rapid population growth and expanded areas of human activities (e.g., cultivation and grazing) [26,39–41]. Xinjiang is one core region of China's "Belt and Road" initiative and plays an important role in business logistics in Europe with Asia. The society and economics in Xinjiang have experienced rapid development in the past two decades. From 2000 to 2018, the population in Xinjiang increased by more than one-third. Meanwhile, the economic growth rate exceeded the average in China [42]. Studying Xinjiang would benefit our understanding of how human activities and climate variation affect local ecosystems inland arid and semi-arid regions. The objectives of this study were to (a) analyze the responses of ecosystem indicators to climate factors and human activities in Xinjiang using satellite-based products and (b) investigate the cause of changes in the ecosystem indicators using the residual analysis approach.

## 2. Materials and Methods

### 2.1. Study Area

Xinjiang (73.66°–96.38°E, 34.42°–49.17°N) in northwestern China has an area of 1.66 million km$^2$ and belongs to the hinterland of the Eurasian continent. The area is locally referred to as "two basins sandwiched by three mountains" (Figure 1). The three mountains are the Altay Mountains, the Tianshan Mountains and the Kunlun Mountains situated from north to south. The two basins are the Junggar Basin located in between the Altai Mountains and the Tianshan Mountains, and the Tarim Basin located in between the Tianshan Mountains and the Kunlun Mountains. Xinjiang has a typical temperate continental climate that has only a few and unevenly distributed precipitation events. For example, annual total precipitation ranges from 100 to 500 mm and annual averaged temperature ranges from 4.0 °C to 8.8 °C in the north of the Tianshan Mountains. By comparison, annual total precipitation ranges from 20 to 100 mm and annual average temperature ranges from 10.0 °C to 13.8 °C in the south of the Tianshan Mountains [41]. The ecological environment in Xinjiang is extremely fragile and sensitive to the climate [26]. The population in Xinjiang is mostly distributed among oases. The population in oases was 3.6 million in 1945 and grew to 21.81 million by 2010, accounting for over 90% of the total population in Xinjiang [9,42]. Water resources largely influence the socio-economic development of Xinjiang, and agricultural activities account for high water consumption [8,39,43].

### 2.2. Data Source and Processing

The data processing roadmap is shown in Figure 2. A summary of all data used in this study is listed in Table S1.

The BESS model dataset (http://environment.snu.ac.kr/, accessed on 1 August 2021) provides time series of global GPP and ET data from 2001 to 2015 at the spatial resolution of 1 km and the temporal resolution of 8 days [44,45]. The BESS model is a process-based biophysical model that uses seven MODIS atmosphere and land products, four other satellite datasets, four reanalysis datasets and three ancillary datasets. The BESS model incorporates atmosphere and canopy radiative transfers, canopy photosynthesis, evapotranspiration and

energy balance processes [44,45]. Some researchers evaluated the performance of the BESS model on a global scale using FLUXNET ground observations, and the results showed that the BESS model outperformed the MODIS products in general [45,46].

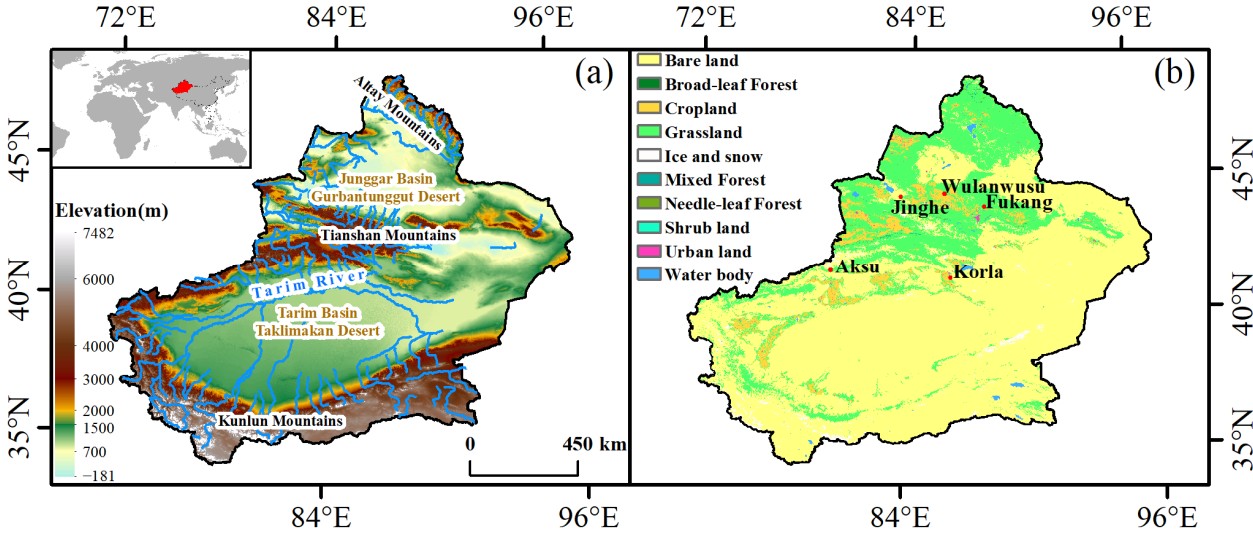

**Figure 1.** Digital elevation model (DEM) (**a**) and the studied observation sites on top of land cover maps in Xinjiang (**b**). Note: The five observation sites are Wulanwusu, Fukang, Jinghe, Korla, and Aksu stations.

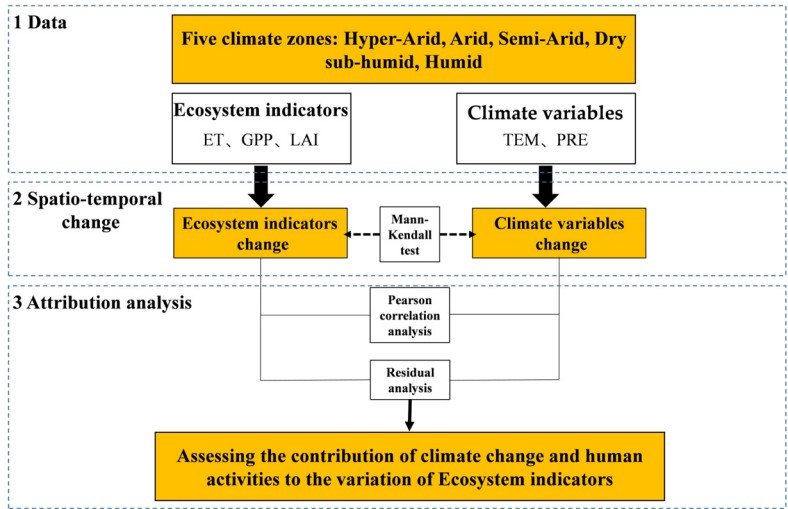

**Figure 2.** Technical flow chart.

The MODIS LAI products were delivered at 500 m spatial resolution and 8-day composite temporal resolution from 2000 to the present (https://ladsweb.modaps.eosdis.nasa.gov/, accessed on 1 August 2021). To be consistent with the BESS dataset resolutions, we resampled the LAI data to the 1 km spatial resolution and used the period spanning from 2001 to 2015. Evaluation studies have found that the MODIS LAI well agreed with the FLUXNET observations at the site scale [47]. The MODIS LAI product has been widely used in studying vegetation phenology, ecosystem productivity and global vegetation variation [48–50]. The ecosystem indicators (ET, GPP, LAI) were cumulated over the entire year.

Daily meteorological data from 66 meteorological stations were acquired from the China Integrated Meteorological Information Sharing System (http://data.cma.cn/, accessed on 1 August 2021). This dataset undergoes strict quality control before data release. Land cover data were downloaded from the European Space Agency (https://cds.climate.copernicus.eu/, accessed on 1 August 2021). DEM data were obtained from

the website of geospatial data cloud (http://www.gscloud.cn/, accessed on 1 August 2021). We used daily temperature and precipitation to calculate mean annual temperature (TEM), and total annual precipitation (PRE) for each year from 2001 to 2015. The studied temperature and precipitation data were interpolated with the spatial resolution of 1 km by the ANUSPLIN software, taking elevation (DEM) as a covariate. The ANUSPLIN has excellent performance in interpolating climate data in Xinjiang [51].

In addition, we collected model validation data, including ET and GPP, from previous publications [52–54]. Monthly GPP data in 2009, 2010, 2012, and 2013 were measured by an eddy covariance system in a typical cropland in Wulanwusu Town (44°17′N, 85°49′E) [52]. The 16-day composited ET and GPP from 2007 to 2009 were collected using an eddy covariance system in a shrubland in Fukang County (44°17′30″N, 87°56′16″E) [53]. The 16-day composited ET data in 2013 were measured by a micro lysimeter in an alpine meadow in Aksu (41°42′N, 80°10′E) [54]. The 8-day composited ET data during 2015 were measured by an eddy covariance system in a cropland in Korla (41°36′N, 86°12′E) [55]. The 10-day composited ET data in 2015 were measured by an eddy covariance system in a shrubland in Jinghe (44°37′N, 83°33′E) [56]. Aridity is generally expressed as a function of precipitation, potential evapotranspiration and temperature. The UNEP proposed a method to classify the climate zones based on the indicator metrics of the aridity index (AI). The AI dataset (https://cgiarcsi.community/, accessed on 1 August 2021) produced by Trabucco and Zomer was used in this study to classify areas into five climate zones in Xinjiang (Figure 3) [57]. The proportion of areas in each climate zone to the total area was 36.9%, 50.2%, 10.1%, 1.4%, and 1.4% for hyper-arid, arid, semi-arid, dry sub-humid, and humid zones, respectively. Areas such as the Gobi Desert were considered as non-vegetated areas and were excluded from analysis. The proportion of vegetated areas in each climate zone to the total area was 8%, 59%, 27.7%, 3.3%, and 2% in the hyper-arid, arid, semi-arid, dry sub-humid, and humid zones, respectively.

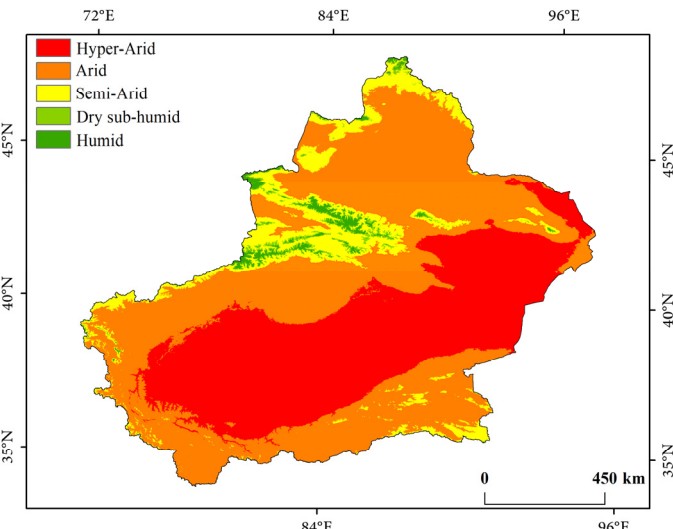

**Figure 3.** The classification map of climate zones based on the aridity index in Xinjiang.

### 2.3. Methods

### 2.3.1. Trend and Correlation Analysis

The Theil–Sen median method was used to calculate the average rate of change in ecosystem indicators and climate factors on an annual basis in Xinjiang from 2001 to 2015 [58]. The Theil–Sen median method is a robust trend method with non-parametric statistics, and it can effectively reduce the effect of outliers. The formula is expressed as follows:

$$\beta = median\left(\frac{x_j - x_i}{j - i}\right), 2001 \leq i < j \leq 2015 \tag{1}$$

where $\beta$ represents the trend of an ecological indicator; $x_i$ and $x_j$ represent the annual values of the ecosystem indicators in the $i$th and $j$th years, respectively. The ecological indicator has an increasing trend if $\beta > 0$; otherwise, the ecological indicator has a decreasing trend.

The Mann–Kendall method is a non-parametric statistical test used to determine the significance of time series trends. This method does not require samples to follow a certain distribution and could mitigate the effects of data noise. The Mann–Kendall method is widely used in time series analysis of hydrology and meteorology.

In the Mann–Kendall method, for a group of time series, the standardized test statistic $Z$ is as follows:

$$Z = \begin{cases} \dfrac{S-1}{\sqrt{var(s)}} & S > 0 \\ 0 & S = 0 \\ \dfrac{S+1}{\sqrt{var(s)}} & S < 0 \end{cases} \tag{2}$$

where

$$S = \sum_{i=1}^{n-1} \sum_{j=i+1}^{n} sgn(x_j - x_i) \tag{3}$$

$$var(s) = \frac{n(n-1)(2n+5)}{18} \tag{4}$$

and

$$sgn(x_j - x_i) = \begin{cases} 1 & x_j - x_i > 0 \\ 0 & x_j - x_i = 0 \\ -1 & x_j - x_i < 0 \end{cases} \tag{5}$$

where $x_i$ and $x_j$ represent the ecological indicator values in years $i$ and $j$, respectively, and $n$ represents the length of the time series. $sgn$ is a sign function. Temporal trends are deemed as significant changes at the 0.05 level when $|Z| > 1.96$. The trends for ecosystem indicators were categorized as significant increasing trends when $\beta > 0$ and $|Z| > 1.96$, significant decreasing trends when $\beta < 0$ and $|Z| > 1.96$, and no significant trends when $|Z| \leq 1.96$.

To further understand the correlations between the studied ecosystem indicators and driven factors on an annual basis, Pearson's correlation coefficient was applied for analysis. This method is widely used in ecological studies [28,59], and the correlation coefficients are calculated as follows:

$$R_{xy} = \frac{\sum_{i=1}^{n}[(x_i - \bar{x})(y_i - \bar{y})]}{\sqrt{\sum_{i=1}^{n}(x_i - \bar{x})^2 \sum_{i=1}^{n}(y_i - \bar{y})^2}} \tag{6}$$

where $x_i$ represents the value of an ecological indicator in the $i$th year; $y_i$ is the annual value of an assembled yearly climate factor (e.g., precipitation or temperature) in the ith year; $n$ is the study period, from 2001 to 2015; $\bar{x}$ and $\bar{y}$ represent the average of $x_i$ and $y_i$, respectively.

### 2.3.2. Residual Analysis

The residual analysis method, proposed by Evans and Geerken [60], is used to distinguish the impacts of climate change and human activities on the long-term change in annual ecosystem indicators in 2001–2015. We chose precipitation and temperature as the key climate factors to deduce the responses of ecosystem indicators to human activities and climate change because precipitation and temperature are two important factors affecting vegetation activities in the region [7,25,26,35]. By establishing the regression model of ecosystem indicators (ET, GPP, and LAI), precipitation, and temperature, the contribution of precipitation and temperature to ecosystem indicators can be predicted grid by grid. The contribution of human activities can be represented by the change in residuals (hereafter referred to as *Vegetation*$_{res}$), which is the difference between the predicted (hereafter referred to as *Vegetation*$_{pre}$) and observed (hereafter referred to as *Vegetation*$_{obs}$) vegetation growth.

$$Vegetation_{pre} = A * PRE + B * TEM + C \tag{7}$$

$$Vegetation_{res} = Vegetation_{obs} - Vegetation_{pre} \tag{8}$$

where *A* and *B* are the regression coefficients and *C* is a constant. $Vegetation_{obs}$ denotes the observed values for ecosystem indicators. $Vegetation_{res} > 0$ indicates that human activities have positive effects on promoting vegetation growth, $Vegetation_{res} < 0$ indicates the opposite, and $Vegetation_{res} \approx 0$ indicates that human activities have weak impacts on vegetation growth. The residual sequence of the study area is analyzed using the linear trend method as follows:

$$slope = \frac{n * \sum_{i=1}^{n}(i * Vegetation) - \sum_{i=1}^{n} i \sum_{i=1}^{n} Vegetation}{n * \sum_{i=1}^{n} i^2 - \left(\sum_{i=1}^{n} i\right)^2} \tag{9}$$

where *n* is the length of the data time series, and *i* is an integer ranging from 1 to *n*. The slope represents the trend of the ecosystem indicators in 2001–2015. The relative contribution of climate change and human activities to ecosystem indicators were calculated according to Table 1 [33].

**Table 1.** Methods for assessing the contribution of climate change and human activities to the variation in vegetation indicators.

| Vegetation Trend | Slope ($Vegetation_{pre}$) | Slope ($Vegetation_{res}$) | Relative Contribution of Climate Change (%) | Relative Contribution of Human Activities (%) | Drivers |
|---|---|---|---|---|---|
| | >0 | >0 | $\frac{slope_{pes}}{slope_{obs}} * 100$ | $\frac{slope_{res}}{slope_{obs}} * 100$ | CC and HA |
| increase | >0 | <0 | 100 | 0 | CC |
| | <0 | >0 | 0 | 100 | HA |
| | <0 | <0 | $\frac{slope_{pre}}{slope_{obs}} * 100$ | $\frac{slope_{res}}{slope_{obs}} * 100$ | CC and HA |
| decrease | <0 | >0 | 100 | 0 | CC |
| | >0 | <0 | 0 | 100 | HA |

Note: CC indicates that the changes in ecosystem indicators were due to climate change; HA indicates that the changes in ecosystem indicators were due to human activities.

## 3. Results

### 3.1. Spatial Variation in Ecosystem Indicators and Climate Variables

The spatial variation trends of ET, GPP, and LAI in Xinjiang from 2001 to 2015 are shown in Figure 4. The trends of ET, GPP, and LAI in different climate zones of Xinjiang are shown in Figure S1 and Table S2. Overall, ET decreased in 73% of the Xinjiang region, and 21% of areas have significant decreasing trends ($p < 0.05$). GPP increased in 56% of the Xinjiang region, and 19% of areas have significant increasing trends ($p < 0.05$). LAI increased in 59% of the Xinjiang region, and 20% areas have significant increasing trends ($p < 0.05$). Three metrics, including ET, GPP, and LAI, all showed increasing trends in the hyper-arid zone, with significant increases in 10%, 35%, and 53% of the regions, respectively. ET showed a decreasing trend in the arid zone, and had significant decreasing trends over 16% of the total area. GPP and LAI showed an increasing trend in the arid zone, and have significant increasing trends over 24% and 27%, respectively, of the total area. ET, GPP, and LAI decreased in the semi-arid zone, and have significant decreasing trends over 34%, 10%, and 8% of the total area, respectively. ET and GPP showed decreasing trends in the dry sub-humid and humid zones, and LAI did not have significant changes in neither dry sub-humid nor humid zones.

The spatial variation trends of PRE and TEM in Xinjiang from 2001 to 2015 are shown in Figure 5. The trends of PRE and TEM in different climate zones of Xinjiang are shown in Figure S2. Overall, PRE increased in 61.2% of the entire Xinjiang region. TEM increased in 50.2% of the entire Xinjiang region. PRE showed increasing trends in the hyper-arid, arid, and semi-arid zones, with increases in 84.1%, 63.1%, and 60% of the regions, respectively. PRE decreased in the dry sub-humid and humid zones, with decreases in 57.6% and 63.3% of the total area, respectively. TEM showed increasing trends in the hyper-arid and arid zones, with increases in 56.9% and 66.1% of the regions, respectively. TEM decreased in the

semi-arid, dry sub-humid, and humid zones, with decreases in 79.3%, 95.7%, and 99.6% of the total area, respectively.

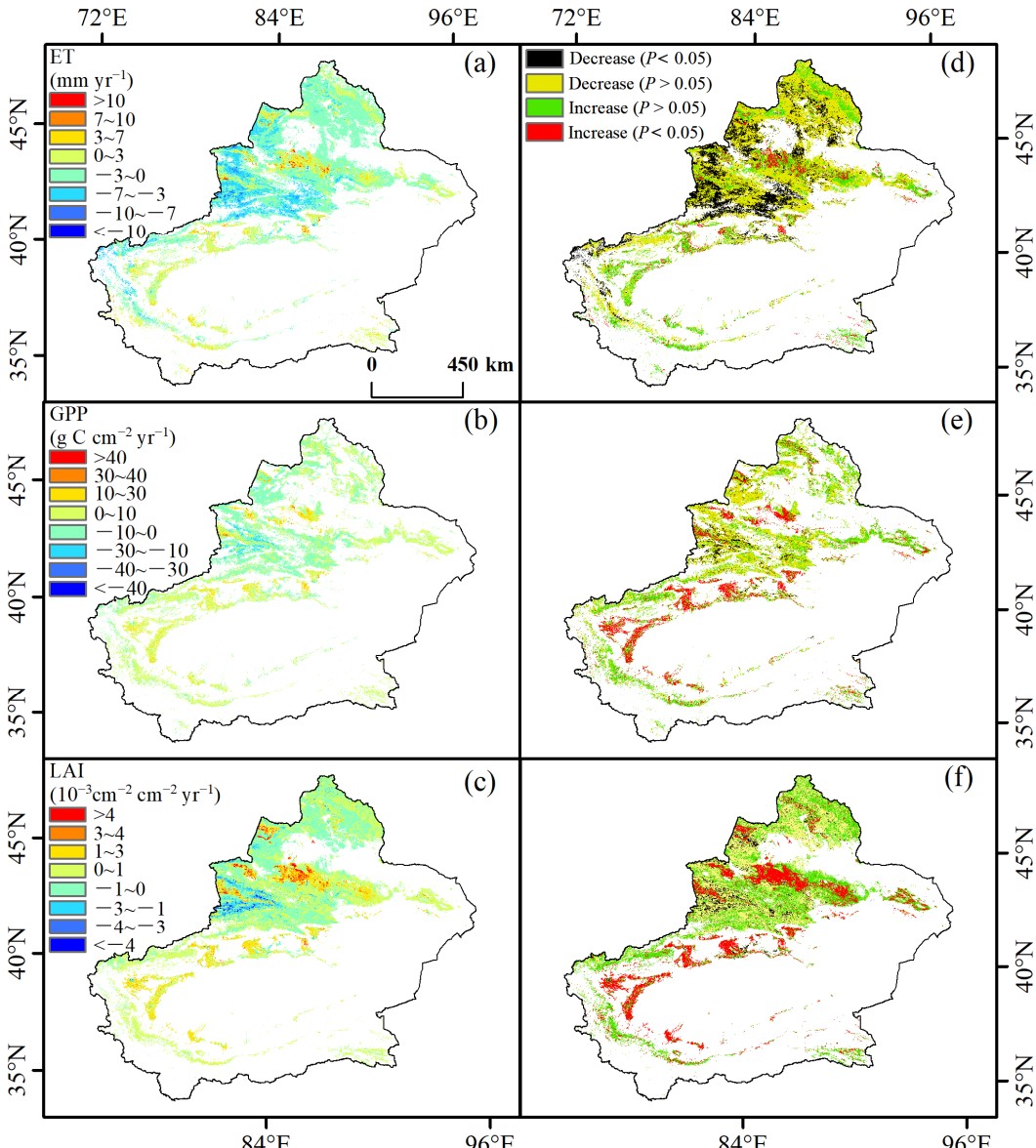

**Figure 4.** The maps show the results for the Theil–Sen median trend of (**a**) ET, (**b**) GPP, (**c**) LAI and the Mann–Kendall test of (**d**) ET, (**e**) GPP, (**f**) LAI from 2001 to 2015.

### 3.2. Temporal Variation in Ecosystem Indicators and Climate Variables in Different Climate Zones

Figure 6 shows the temporal variation in ET. There was no obvious change in ET in the entire Xinjiang region, and ET had decreasing trends before 2009 and increasing trends after 2009. ET reached its lowest value in 2008 due to a severe drought in the corresponding year (Figure S3). The trend of the changes in ET in the hyper-arid and arid zones was the same as in the entire Xinjiang region, and their annual ET values are lower than that of the entire Xinjiang region. ET decreased significantly, with the rate of 2.16 and 1.9 mm yr$^{-1}$ ($p < 0.01$) in the semi-arid and dry sub-humid zones, respectively, and their annual ET values are higher than that of the entire Xinjiang region. ET decreased in the humid zone.

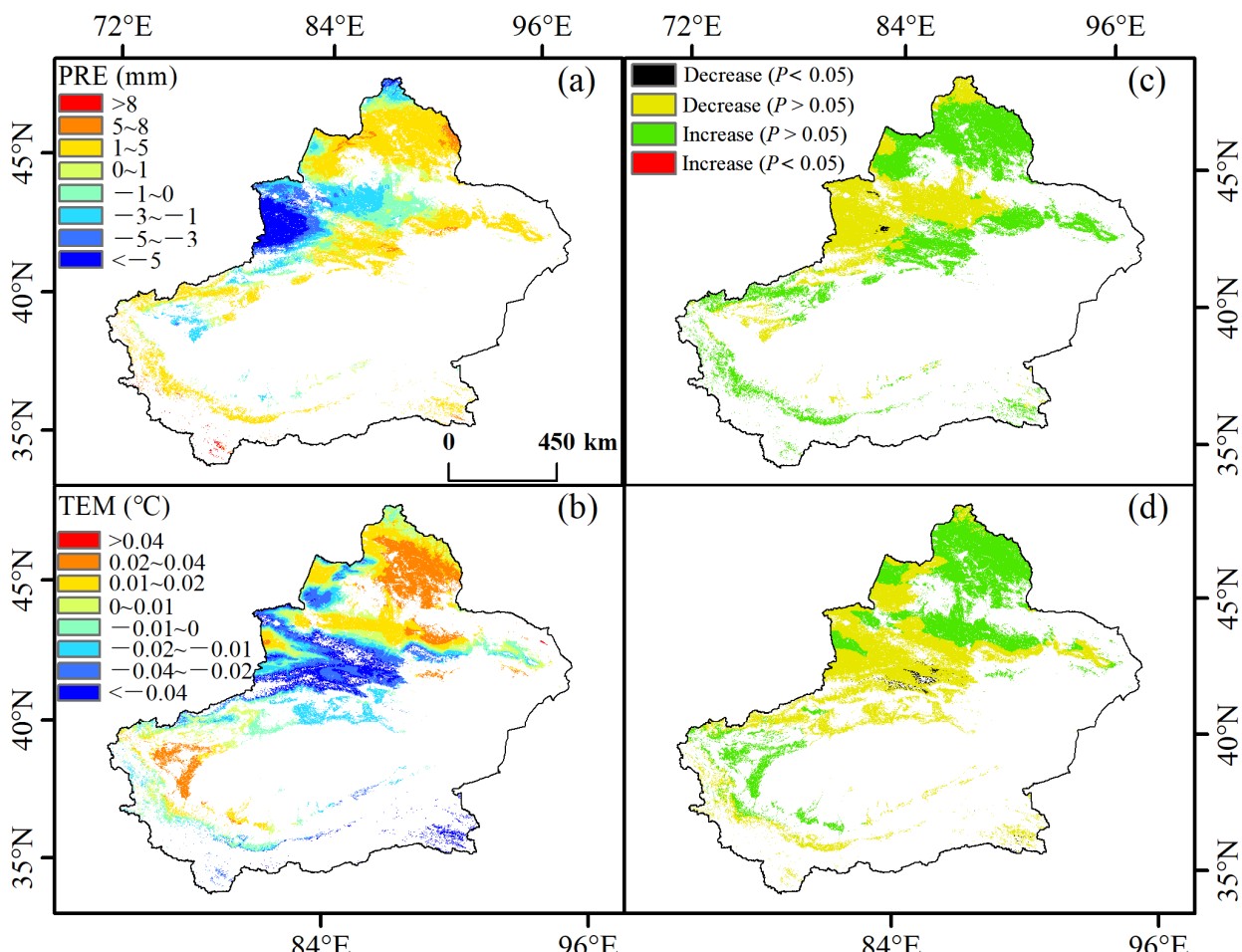

**Figure 5.** The maps show the results for the Theil–Sen median trend of (**a**) PRE, (**b**) TEM and the Mann–Kendall test of (**c**) PRE, (**d**) TEM from 2001 to 2015.

Figure 7 shows the temporal variation in GPP. GPP increased at the rate of 1.19 gC m$^{-2}$ yr$^{-1}$ for the entire Xinjiang region. GPP had a peak value of 322 gC m$^{-2}$ yr$^{-1}$ in 2013. GPP had significant increasing trends at the rates of 3.49 and 2.4 gC m$^{-2}$ yr$^{-1}$ in hyper-arid and arid zones, respectively. GPP in both hyper-arid and arid zones had peak values in 2013 and they were 254 gC m$^{-2}$ yr$^{-1}$ and 279 gC m$^{-2}$ yr$^{-1}$, respectively. Annual average GPP values in both hyper-arid and arid zones are lower than that in the entire Xinjiang region. GPP decreased in the semi-arid, dry sub-humid, and humid zones. Annual average GPP values in the semi-arid, dry sub-humid, and humid zones are higher than that in the entire Xinjiang region. GPP sightly decreased in the humid zone.

Figure 8 shows the temporal variation in LAI. LAI increased at the rate of 0.00027 m$^2$ m$^{-2}$ yr$^{-1}$ ($p < 0.05$) in the entire Xinjiang region. LAI had a peak value of 0.045 m$^2$ m$^{-2}$ in 2013. LAI increased significantly with the trend of 0.00074 and 0.00052 m$^2$ m$^{-2}$ yr$^{-1}$ ($p < 0.001$) in the hyper-arid and arid zones, respectively. LAI in both hyper-arid and arid zones had peak values in 2013 and they were 0.046 and 0.040 m$^2$ m$^{-2}$, respectively. LAI showed a weak decreasing trend in the semi-arid zone, with the LAI values fluctuating in the range of 0.051 to 0.063 m$^2$ m$^{-2}$. There was no obvious change in LAI in the dry sub-humid and humid zones.

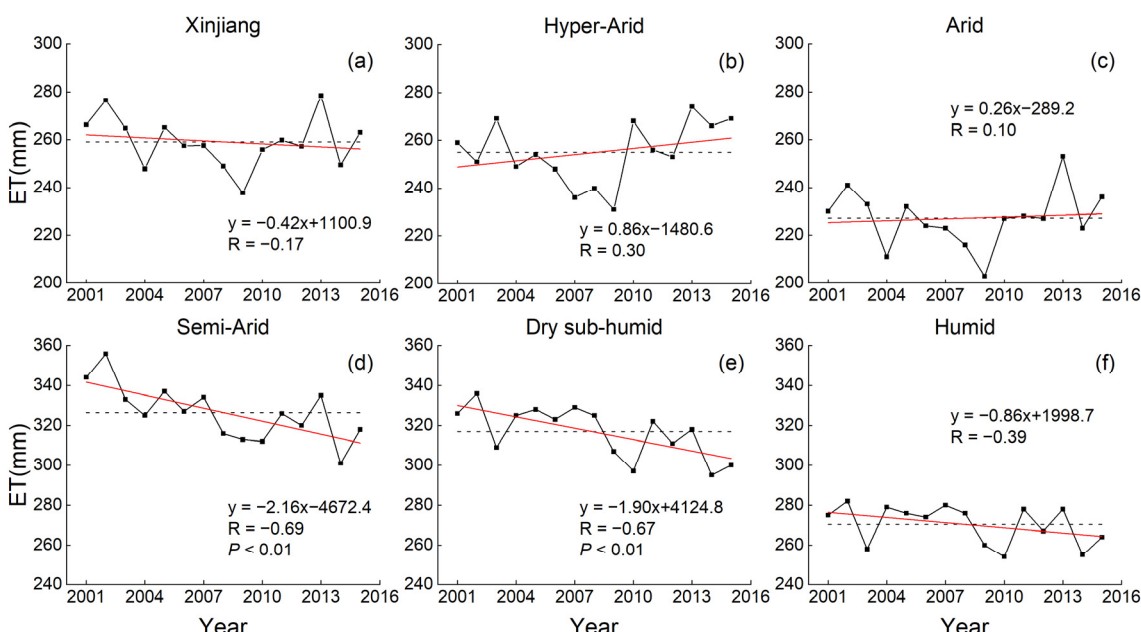

**Figure 6.** Inter-annual variations in ET in different climate zones from 2001 to 2015 are shown for (**a**) the entire Xinjiang region, (**b**) hyper-arid region, (**c**) arid region, (**d**) semi-arid region, (**e**) dry sub-humid region, and (**f**) humid region.

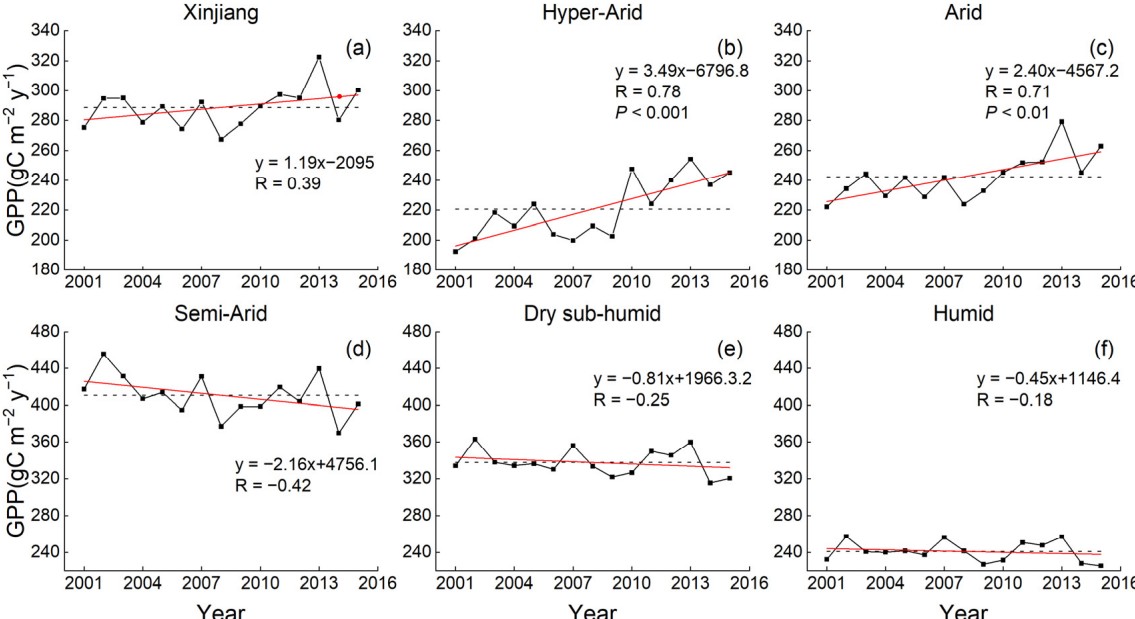

**Figure 7.** Inter-annual variations in GPP in different climate zones from 2001 to 2015 are shown for (**a**) the entire Xinjiang region, (**b**) hyper-arid region, (**c**) arid region, (**d**) semi-arid region, (**e**) dry sub-humid region, and (**f**) humid region.

Figure 9 shows the temporal variation in PRE. PRE increased at the rate of 0.62 mm yr$^{-1}$ in the entire Xinjiang region. In 2007, PRE reached a peak value of 333.6 mm. The hyper-arid and arid zones had the same trend in PRE changes as the entire Xinjiang region, and their annual PRE values were lower than that in the entire Xinjiang region. PRE had increasing trends at the rates of 1.47, 0.6, and 0.37 mm yr$^{-1}$ in the hyper-arid, arid, and semi-arid zones, respectively. PRE decreased weakly, with the rates of 0.11 and 0.25 mm yr$^{-1}$ in the dry sub-humid and humid zones, respectively, and their annual PRE values are higher than that of the entire Xinjiang region.

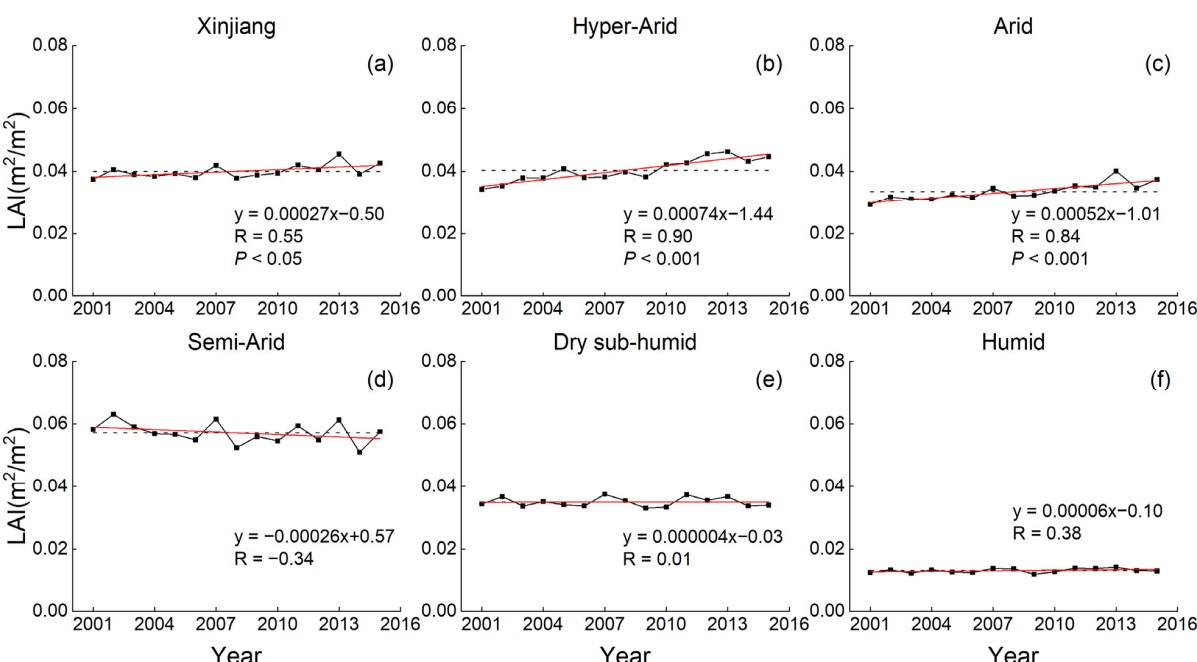

**Figure 8.** Inter-annual variations in LAI in different climate zones from 2001 to 2015 are shown for (**a**) the entire Xinjiang region, (**b**) hyper-arid region, (**c**) arid region, (**d**) semi-arid region, (**e**) dry sub-humid region, and (**f**) humid region.

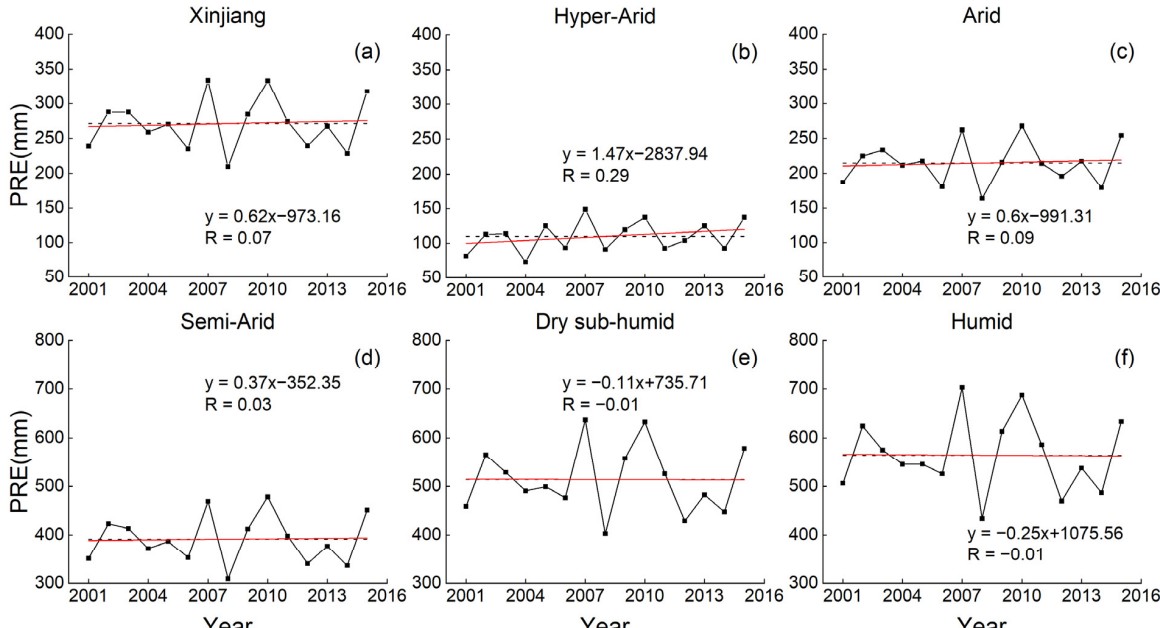

**Figure 9.** Inter-annual variations in PRE in different climate zones from 2001 to 2015 are shown for (**a**) the entire Xinjiang region, (**b**) hyper-arid region, (**c**) arid region, (**d**) semi-arid region, (**e**) dry sub-humid region, and (**f**) humid region.

Figure 10 shows the temporal variation in TEM. TEM decreased weakly, with the rate of $0.0059\ °C\ yr^{-1}$ in the entire Xinjiang region. In 2007, the TEM reached a peak value of $5.4\ °C$. In the hyper-arid and arid zones, TEM increased trends at the rates of 0.0008 and $0.0018\ °C\ yr^{-1}$, respectively. Their annual TEM values are higher than that in the entire Xinjiang region. Moreover, in the semi-arid, dry sub-humid, and humid zones, TEM showed a weak decreasing trend with rates of 0.02, 0.0352 and $0.0359\ °C\ yr^{-1}$, respectively.

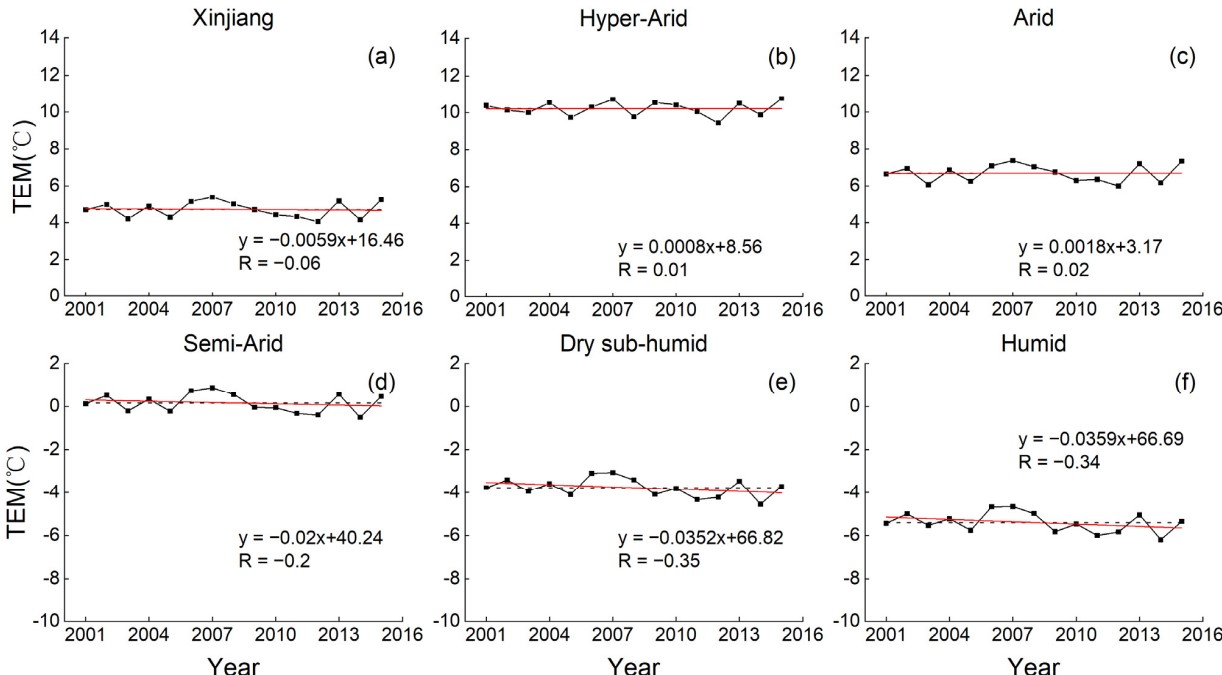

**Figure 10.** Inter-annual variations in TEM in different climate zones from 2001 to 2015 are shown for (**a**) the entire Xinjiang region, (**b**) hyper-arid region, (**c**) arid region, (**d**) semi-arid region, (**e**) dry sub-humid region, and (**f**) humid region.

### 3.3. Correlation between Ecosystem Indicators and Climate Parameters

The correlation coefficients between the ecosystem indicators (ET, GPP, and LAI) and climate factors (PRE and TEM) during the period 2001–2015 are shown in Figure 11. These correlations in different climate zones of Xinjiang are shown in Figure S4 and Table S3. The correlation coefficients vary considerably across different climate zones. The annual precipitation was positively correlated with ET, GPP, and LAI in 70.5%, 75.8%, and 68.4% of the entire Xinjiang region, respectively, among which 10.9%, 12.4%, and 9.9% were significantly positively correlated.

The positive correlation areas between precipitation and the ecosystem indicators increased first and then decreased as the climate zones became wetter. The negative effect of precipitation on ET was greater than the positive effect in the dry sub-humid and humid zones. The negative effect of precipitation on ET was greater than 50% of the total area in the dry sub-humid and humid zones. The positive correlation areas between precipitation and ET and GPP was greatest in the arid zone, with both being above 77%, and the areas of maximum significant correlation were in the hyper-arid (17%) and arid zones (13.8%). The positive correlation areas between precipitation and LAI were greatest in the semi-arid zone, with an area of 72%. The area of maximum significant correlation was in the arid zone, with an area of 16.2%.

The correlation between temperature and the ecosystem indicators is lower than that between precipitation and the ecosystem indicators in the entire Xinjiang region. The mean annual temperature was positively correlated with ET, GPP, and LAI in 68.2%, 53%, and 63.5% of the entire Xinjiang region, respectively, among which 4.3%, 2.4%, and 5.6% were significantly positively correlated. The positive correlation areas between temperature and ET increased as the climate zones became wetter; the maximum positive correlation areas were 94.8% in the humid zone, with 23% of the areas having a significant positive correlation. The positive correlation areas between temperature and GPP increased as the climate zones became wetter; however, the negative correlation was predominant in the hyper-arid and arid zones, with 56.1% and 53.4% of the negative correlation areas, respectively. The positive correlation areas between temperature and LAI increased first and then decreased as the climate zones became wetter. The negative correlation was

dominant in the hyper-arid zone, with 58.2% of the negative correlation areas, and the positive correlation was predominant in other climate zones. The largest positive correlation areas were 70.3% in the semi-arid zone.

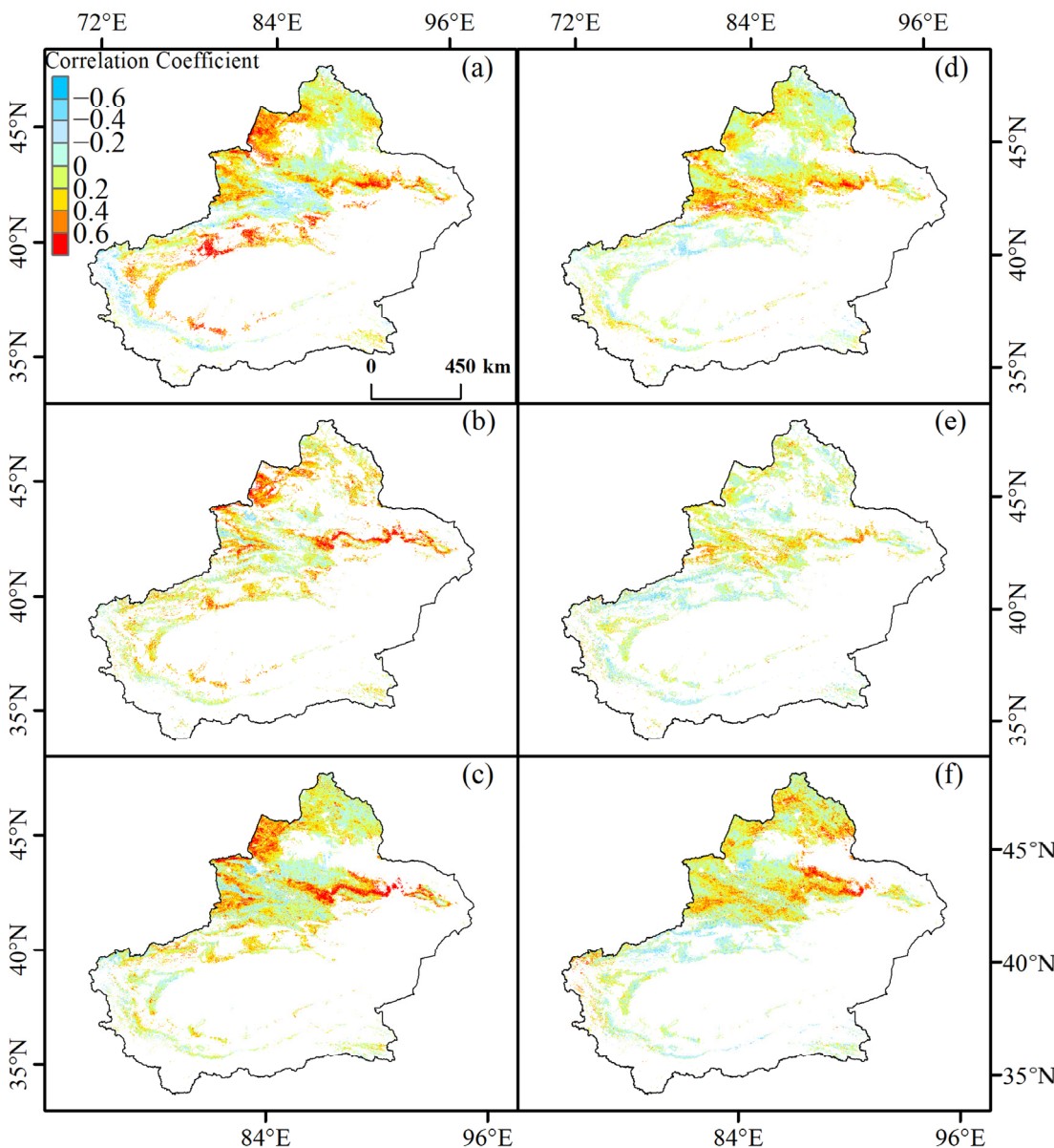

**Figure 11.** Spatial pattern of the correlation between precipitation, temperature, and ecosystem indicators from 2001 to 2015. (**a**) Precipitation-ET, (**b**) Precipitation-GPP, (**c**) Precipitation-LAI, (**d**) Temperature-ET, (**e**) Temperature-GPP, and (**f**) Temperature-LAI.

### 3.4. Residual Trend Analysis of the Vegetation and Its Related Factors

The mean annual TEM, total annual PRE, and ecosystem indicators were used for residual analysis (RA1). The residual analysis showed that ET, GPP, and LAI responded differently to climate change and human activities, and the relative contributions of climate change and human activities to the changes in the trend of the ecosystem indicators are shown in Figure 12. These relative contributions in different climate zones of Xinjiang are shown in Figure S5 and Table S4. Overall, in Xinjiang, climate change accounted for 19.1%, 23.8%, and 23.2% of the total variation in ET, GPP, and LAI, respectively, while those of human activities are 80.9%, 78.2%, and 76.8%, respectively. The ET, GPP, and LAI trends are obviously dominated by human activities, of which the contributions are greater than 70%.

The contribution of climate change and human activities to the changes in the ecosystem indicators vary considerably across different climate zones.

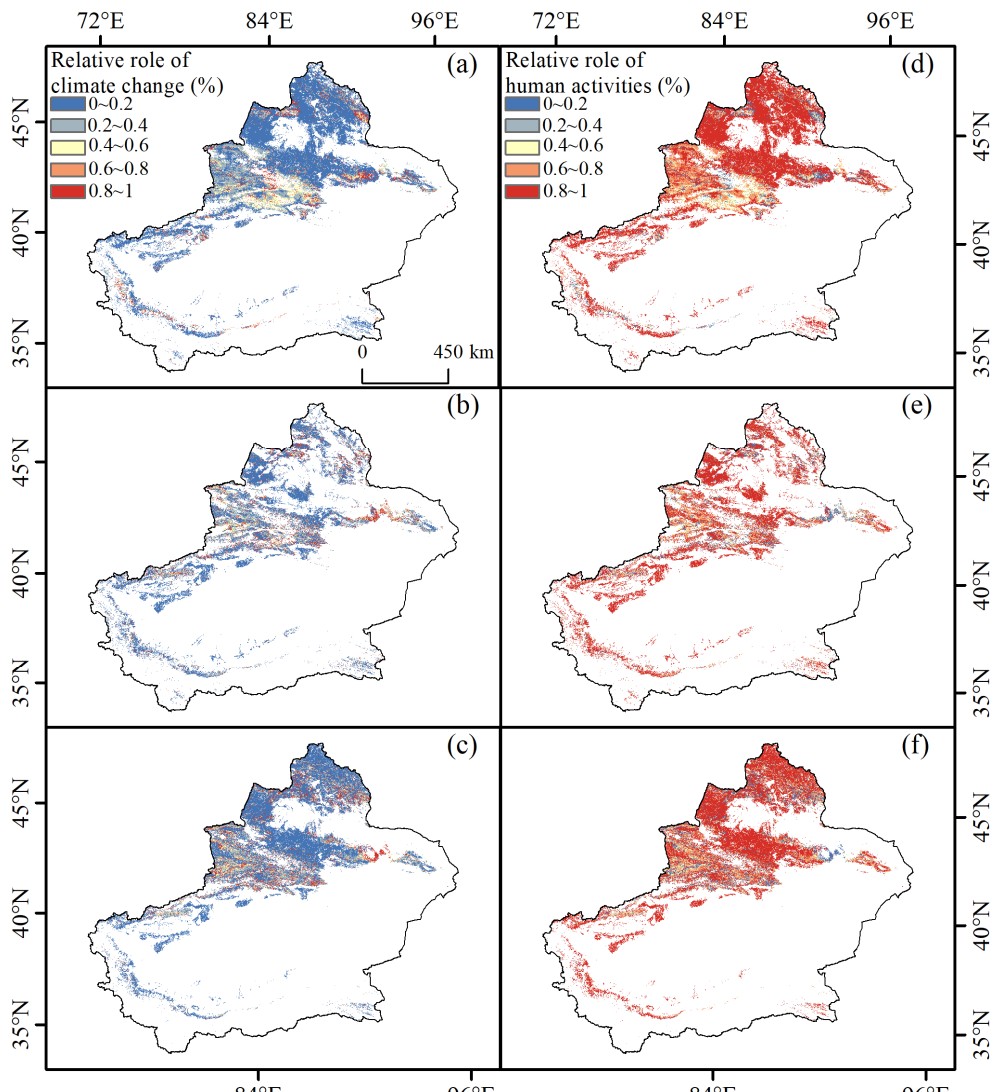

**Figure 12.** Spatial distribution of the relative role of climate change and human activities on the variation in the ecosystem indicators from 2001 to 2015. (**a**) Relative role of climate change in ET, (**b**) Relative role of climate change in GPP, (**c**) Relative role of climate change in LAI, (**d**) Relative role of human activities in ET, (**e**) Relative role of human activities in GPP, and (**f**) Relative role of human activities in LAI.

　　For the ET trends, human activities contributed the most to the changes in the ET trends in the arid zone, accounting for 83.8%, and climate changes contributed to 16.2%. The impact of human activities weakened as the climate became suitable for vegetation growth. There was minimal impact of human activities on ET in the humid zone; the impact of human activities had a relative contribution of 56.2%, while the impact of climate change had a relative contribution of 43.8%.

　　For the GPP trends, human activities contributed the most to the changes in the GPP trends in the hyper-arid zones, accounting for 80.1%, and climate changes contributed to 19.9%. The impact of human activities in the arid and semi-arid zones had relative contributions of 76.3% and 75.4%, while those of climate change were 23.7% and 24.6%, respectively. The impact of human activities weakened in the dry sub-humid and humid zones; the impact of human activities had relative contributions of 68.2% and 69.4%, while those of climate change were 31.8% and 30.6%, respectively.

For the LAI trends, the impact of climate change and human activities on LAI is similar to GPP, but the impact of human activities on LAI is enhanced, especially in the hyper-arid and humid zones. The relative contributions of human activities increased by 2.6% and 6.8% in the hyper-arid and humid zones, respectively, compared with GPP.

### 3.5. Evaluation of the BESS Products in the Xinjiang

In this study, the ecosystem model (BESS) was validated with respect to GPP and ET estimates using satellite-derived data (MODIS products) and field data collected through eddy covariance flux sites and the data from previous publications in Xinjiang. Figure S6 compares satellite-derived data with BESS GPP and ET on the multi-year average values for each pixel. There are good agreements between the BESS-simulated and satellite-derived data, indicated by $R^2 = 0.67$ and RMSE = 71.16 mm yr$^{-1}$ for ET, and $R^2 = 0.76$ and RMSE = 107.04 gC m$^{-2}$ yr$^{-1}$ for GPP. The results show that the BESS model performed fairly well in estimating ET, with $R^2$ of 0.68 ($p < 0.001$), 0.88 ($p < 0.001$), 0.74 ($p < 0.001$), and 0.62 ($p < 0.001$) and RMSE of 8.37.5.97, 1.37, and 3.5 for Aksu (alpine meadow), Fukang (shrubland), Jinghe (shrubland), and Korla (cropland), respectively. This model also exhibited excellent performance in estimating ET in the Yellow River Basin, China (irrigated cropland) [61]. Compared with the MODIS ET products in the Aksu, Fukang, and the Yellow River Basin, the BESS ET performed better (Figure 13) [61].

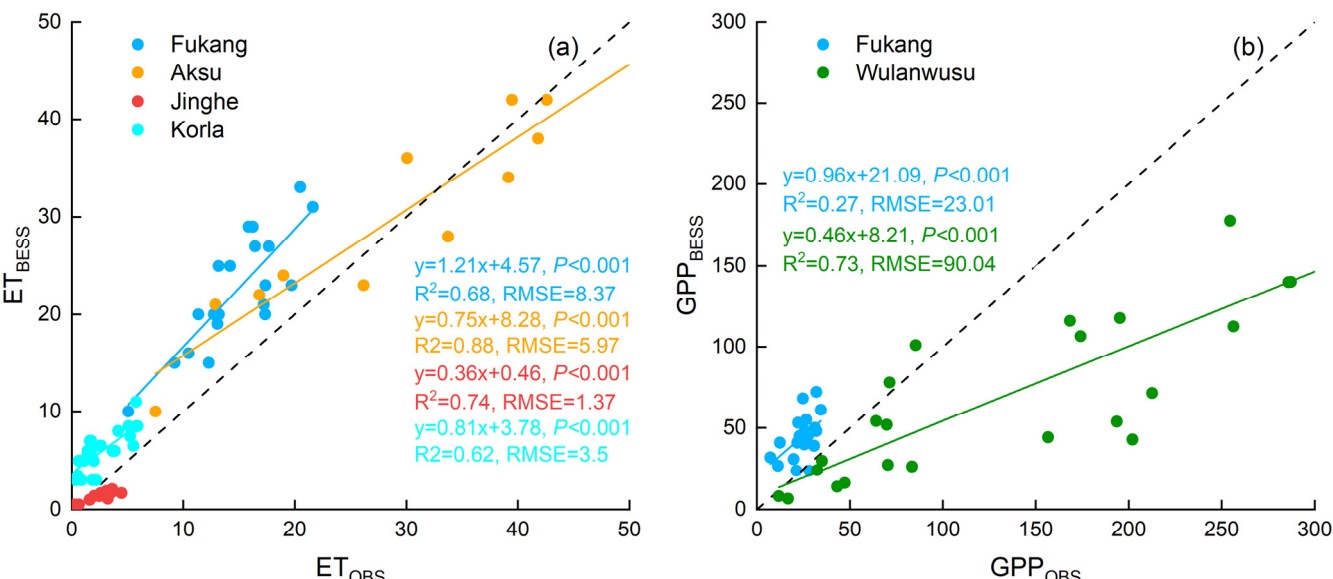

**Figure 13.** Relationship between observed data and the BESS model in Xinjiang. (**a**) The comparison between ET$_{OBS}$ and ET$_{BESS}$; (**b**) the comparison between GPP$_{OBS}$ and GPP$_{BESS}$.

The result showed that the BESS GPP performed poorly in Fukang (shrubland); the result explained only 27% of the variation in the observed GPP, and the BESS GPP over-estimated the GPP, with $R^2$, slope, and RMSE values of 0.27, 0.96, and 23.01 g C m$^{-2}$ 16 days$^{-1}$, respectively (Figure 13). The BESS GPP performed fairly well in Wulanwusu (cropland), with $R^2$, slope, and RMSE values of 0.73, 0.46, and 90.04 g C m$^{-2}$ month$^{-1}$, respectively (Figure 13). However, the model performance was still better than MODIS GPP in Fukang ($R^2 = 0.16$, slope = 0.45, RMSE = 11.01 g C m$^{-2}$ 16 days$^{-1}$) [53]. The BESS model ($R^2 = 0.72$) performed better than the MODIS product ($R^2 = 0.0014$) in estimating GPP in the grassland areas of China, such as Changling (44°59′N, 123°51′E), Dangxiong (30°05′N, 91°07′E), Haibei (37°37′N, 101°18′E), and the Inner Mongolia station (43°55′N, 116°68′E) [62].

## 4. Discussion

The response of vegetation to climate factors is key to explaining the ecosystem dynamics and structure [8,9,11,63]. Climate change has a different influence on the ecosystem indicators across different climate zones. Arid and semi-arid zones dominate the Xinjiang region, and therefore, water availability is the main climate factor for limiting vegetation growth [8,11,64]. Precipitation is positively correlated with the ecosystem indicators in more than 68% of the area due to the vulnerability of vegetation growth to water variation (Figure 11). The positive correlation between the climate factors and the ecosystem indicators is lower in the hyper-arid zone than in the arid zone. It is possible that soil moisture deficit during the growing season counteracts the benefits of precipitation and that moisture stress reduces the positive effect of the climate factors on the ecosystem indicators [65]. As the climate becomes wetter, the positive effect of precipitation on the ecosystem indicators diminishes due to increased precipitation (Figures 9 and 11). The negative effect of precipitation on ET and LAI was more than 50% in the dry sub-humid and humid zones, indicating that vegetation is rarely affected by water stress because water availability in a wet climate is not controlled by precipitation (Figure 9) [9,40]. Low radiation and temperatures are frequently caused by excessive precipitation, which restricts the development of plants [9]. The correlation between the ecosystem indicators and temperature increases as the climate becomes suitable for vegetation growth (Figures 10 and 11). However, the climate conditions are usually cold and humid due to the low temperature, low evapotranspiration, and high precipitation (Figures 9 and 10) [40]. Vegetation growth is limited by temperature or radiation in the dry sub-humid and humid zones [66], while the negative effect on vegetation growth due to the cooling effect of precipitation reduces the sensitivity of vegetation (LAI) to temperature (Figure 11). The response of local ecosystem indicators to climate factors varies significantly. Precipitation and temperature interact to influence the changes in the ecosystem indicators.

The conditions of the underlying land surface can affect the changes in the ecosystem indicators significantly. All factors such as increased coverage of natural vegetation, decreased forest area, increased arable land, and urbanization will influence ecosystem processes [9,15,67]. Climate change is one of the crucial contributors to the changes in the ecosystem indicators. Human activities significantly modify the ecological environment of Xinjiang. In addition to a large amount of farmland and pasture, ecological water transfer projects in the basin have also been implemented in Xinjiang [68]. Global warming and vegetation greening lead to an accelerated terrestrial carbon–water cycle [36,69]. This study suggested that ET in the entire Xinjiang region presented a decreasing trend, whereas GPP and LAI showed an increasing trend, which is consistent with the findings of previous studies [8,15,20,26,69]. On the one hand, the inconsistent changing trends of ET and both GPP and LAI in Xinjiang may be due to the apparent differences in the sensitivity of ET to climate and vegetation changes in various zones [68]. On the other hand, the carbon cycle and vegetation growth are also affected by human activities [70]. In hyper-arid and arid zones, ecosystem indicators exhibit increasing trends. The improvement in the ecosystem indicators was mainly caused by human activities, accounting for more than 80% and 76% of the ecosystem changes, respectively. The ecosystem indicators improved mainly due to the increased area of oasis irrigation and the ecological water transfer project. The vegetation in this area was distributed primarily in the oasis, where the rapid population growth led to a dramatic expansion of farmland area [39]. From 2001 to 2015, the cultivated area in the Tarim River Basin (hyper-arid and arid zones) in southern Xinjiang increased by 51.6% (Table S5), where natural grassland was the primary source for farmland reclamation [43]. The increased cultivated land is mainly planted with high water consumption cotton. From 2001 to 2015, the cotton sown area increased by 2.6 times (Table S5). Decreasing natural land cover and increasing artificial land were also observed in the Aksu River Basin [71]. The increase in agricultural water use in this area significantly shifted the ecological water use [39,71]. However, ecological water transfer projects, subject to certain limitations, could protect the survival of existing vegetation rather than facilitating the restoration of natural

vegetation effectively [68]. ET ($p < 0.05$), GPP, and LAI presented a decreasing trend in the semi-arid zone that may be caused by the increasing trend of grazing intensity. Human activities contributed to more than 76% of the ecosystem changes. In the effective statistical area of this region based on grassland, grazing was the primary human activity. In recent decades, the grazing intensity has increased [16,41]. Grassland degradation in Xinjiang due to overgrazing has become a common phenomenon [40,41,59], especially in the Ili River Valley, which has the most severe degradation of high-quality grassland [59]. From 2001 to 2015, the number of livestock in the Ili region (semi-arid zone) increased by 5.6% (Table S5). In many grazing areas, LAI decreased due to the consumption of grass by livestock. The reduction in plant transpiration leads to lower ET, which causes an evident decline in grassland productivity [40]. High-intensity grazing seriously inhibits grass growth, which reduces plant transpiration and promotes the production of surface runoff, resulting in a sharp decrease in ET [40]. Grazing at a high intensity can adversely affect grassland and possibly aggravate desertification process [64,72]. The area of dry sub-humid and humid zones in Xinjiang accounts for only about 4% of the area of the entire Xinjiang region. Human activities, with a weakened effect on the ecosystem indicators, are still a dominant factor. Alpine meadow is the primary land cover type, and grassland degradation still affects sub-humid and humid zones.

Xinjiang is a typical fragile and vulnerable ecological area [8,26]. As the ecological environment in Xinjiang is very fragile, the frequency and intensity of sandstorms are increasing with the continuous expansion of the desert areas [72]. Human activities dominate the changes in the ecosystem process of Xinjiang (Figure 11). In hyper-arid and arid zones, LAI and GPP increased significantly, while ET also increased, indicating that the greening of vegetation comes at the cost of higher water consumption [16]. Given the limited water resources in this area, adjusting the structure of agricultural oasis, improving the water use efficiency, and developing water-saving and eco-agriculture are possible solutions [73–75]. For example, planting drought-resistant and salt-tolerant plants such as Chinese wolfberry (*Lycium barbarum* L.) and sea buckthorn (*Hippophae rhamnoides* L.) can not only improve soil structure and facilitate wind break sand fixation but also increase farm income and achieve the sustainable development of oasis agriculture [74,75]. The semi-arid zone climate has the most significant positive effect on LAI. As human activities have led to grassland degradation, we should pay attention to protecting natural vegetation and grassland. Protective measures, such as reducing livestock density and implementing fenced grazing, should be implemented in pastoral areas to mitigate the impact of grazing on grassland degradation [40,41,72]. In dry sub-humid and humid zones, the effect of human activities is relatively small due to the high altitude. However, we still need to pay attention to protecting natural vegetation.

In dry land, precipitation played a significant role in determining vegetation growth [60]. Some previous studies have ignored the importance of precipitation period on vegetation changes [25,30]. This shows that it is important to evaluate several different periods of precipitation accumulation in order to remove the greatest impact of precipitation on ecosystem indicators. Therefore, correlations are calculated for many different combinations of monthly precipitation accumulation for each ecosystem indicators pixel, allowing for the identification of the pixel's distinct optimum correlation [60]. Ecosystem indicators are more correlated with accumulating precipitation between April and August than in other precipitation accumulation periods in the entire Xinjiang region (Table S6). Correlation matrices similar to that in Table S6 are produced for each pixel in the entire Xinjiang region for ecosystem indicators (Table S7). In April and August, there is a stronger correlation between ecosystem indicators and accumulating precipitation period for each pixel, which is consistent with the entire Xinjiang region. We established a multiple linear regression model among the annual ecosystem indicators, annual TEM, and best accumulation precipitation and then performed residual analysis (RA2). Overall, RA2 produces similar results to RA1 (Figures 14 and S7). In general, all two versions predict that human activities were the dominant cause of the change in ecosystem indicators. However, in RA2, the

impact of human activities on the ecosystem indicators was reduced by about 4% when compared with RA1. This is also to be expected as using the optimum precipitation periods means that the largest amount of variation in the ecosystem indicators is removed by the precipitation association. The above comparison demonstrates the importance of the accumulation precipitation period to ecosystem indicators.

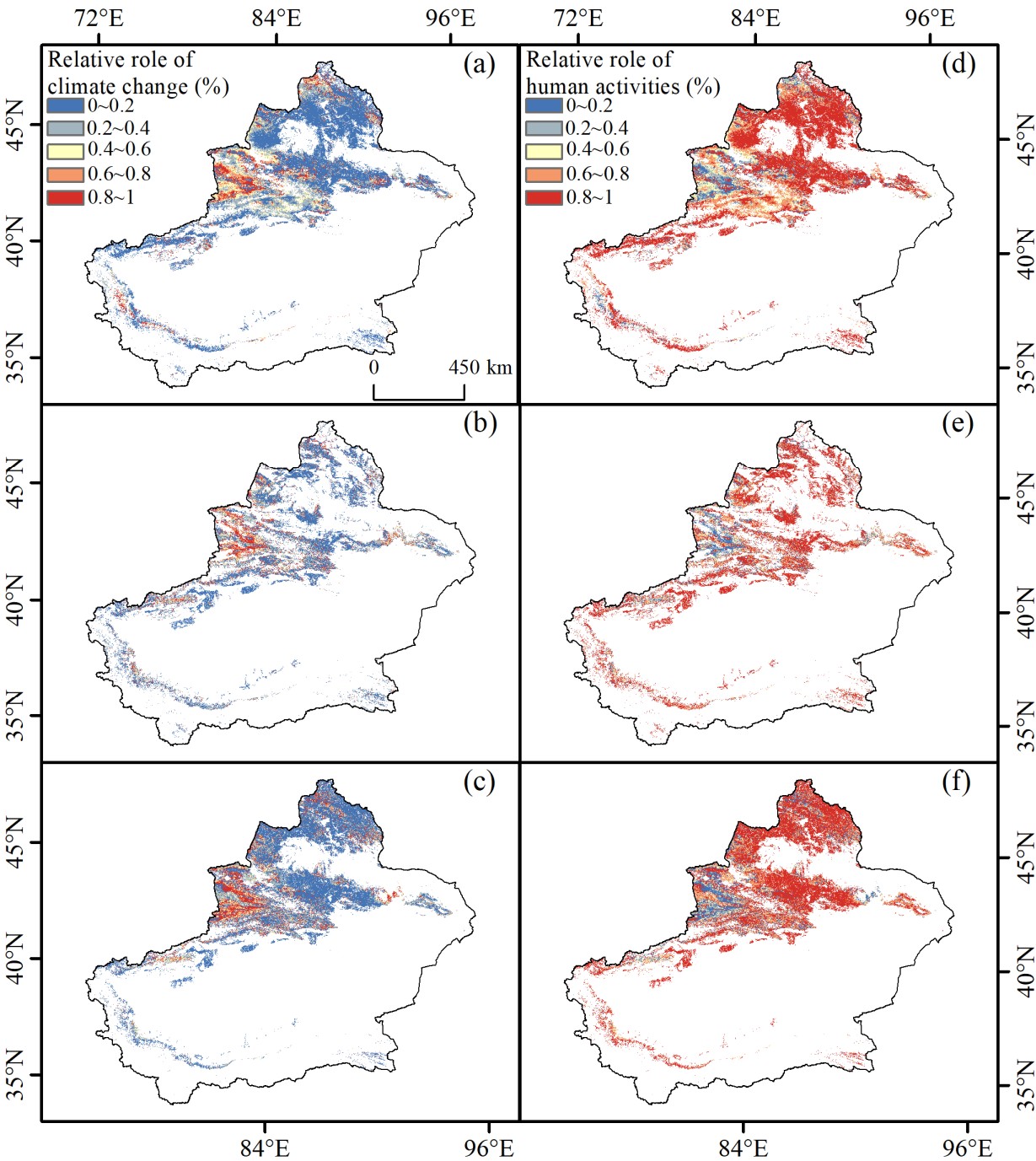

**Figure 14.** Spatial distribution of the relative role of climate change and human activities on the variation in the ecosystem indicators from 2001 to 2015. (**a**) Relative role of climate change in ET, (**b**) Relative role of climate change in GPP, (**c**) Relative role of climate change in LAI, (**d**) Relative role of human activities in ET, (**e**) Relative role of human activities in GPP, and (**f**) Relative role of human activities in LAI. Note: PRE used the optimum precipitation accumulation period.

In this study, we studied three main ecosystem indicators, namely, ET, GPP, and LAI. These indicators might not help comprehensively assess the impacts of climate change and human activities on ecosystem functions and conditions—there is a need to include more indicators or synthesized indices for comprehensive analysis. We only tested two main climate variables (i.e., temperature and precipitation) to quantify the contributions of climate variability on vegetation growth. There are uncertainties in quantifying the relative contributions of climate change and human activities on affecting ecosystems. In the future, we can consider using multiple climate variables and different methods to understand the impacts of climate variation and human activities on ecosystems. The BESS products were used for analysis because they have improved data quality and considered the effects of sunlit and shade canopy on GPP and ET as compared with the current MODIS products. The BESS model still has limitations and uncertainties; for example, it could underestimate GPP for the croplands and it relies on remote sensing data. For comprehensive analysis on future scenarios, we might need to analysis results generated from complex land surface models (such as the Community Land Model).

### 5. Conclusions

The objective of this study was to analyze the response of ecosystem indicators to climate change and human activities across different climate zones in Xinjiang. The assessment was performed using the GPP, ET, and LAI from 2001 to 2015. In general, the positive correlation of ecosystem indicators with precipitation was higher than the correlation with temperature. Precipitation improved ecosystem indicators in the hyper-arid zone, and temperature had negative impacts on GPP and LAI in the hyper-arid zone, suggesting that water availability is a limiting factor in the hyper-arid zone. As the climate becomes more suitable for vegetation growth, the positive effects of temperature increase and the negative effects of precipitation increase. GPP and LAI increased and ET decreased in 2001–2015 in Xinjiang. Climate change and human activities were both driving factors affecting ecosystem indicators. Among all ecosystem indicators change that occurred in Xinjiang, the relative contributions of climate change accounted for 19%, 23.8%, and 23.2% in ET, GPP, and LAI, and the relative contributions of human activities were responsible for 80.9%, 78.2%, and 76.8%, respectively, indicating that human activities were the dominant driver of changes in ecosystem indicators. The relative contributions of human activities have considerable impacts on vegetation indicators in the hyper-arid and arid zones. The results from this study highlight the impact of human activities on ecosystem change. Research on changes in ecosystem indicators in arid areas must consider the impact of human activities, such as grazing, irrigation, and land use changes.

**Supplementary Materials:** The following supporting information can be downloaded at: https://www.mdpi.com/article/10.3390/rs14163911/s1, Figure S1: The Sen trend (a) and the Mann-Kendall test (b) of ET, GPP, and LAI in different climate zones from 2001 to 2015; Figure S2: The Sen trend of PRE and TEM in different climate zones from 2001 to 2015; Figure S3: The drought affected areas in Xinjiang from 2001 to 2015; Figure S4: Spatial pattern of the correlation between precipitation (a), temperature (b), and ecosystem indicators (ET, GPP, and LAI) in different climate zones from 2001 to 2015; Figure S5: Spatial distribution of the relative role of climate change (a) and human activities (b) on the variation of the ecosystem indicators (ET, GPP, and LAI) in different climate zones from 2001 to 2015; Figure S6: Relationship between satellite-driven data (MODIS) and the BESS model in Xinjiang; Figure S7: Spatial distribution of the relative role of climate change (a) and human activities (b) on the variation of the ecosystem indicators (ET, GPP, and LAI) in different climate zones during the optimum precipitation accumulation period from 2001 to 2015; Table S1: Summary of the data; Table S2: Percentage of areas with Inter-annual variations of ecosystem indicators; Table S3: Percentage of areas with the positive correlation between ecosystem indicators and climate variables; Table S4: The contribution of climate change and human activities to the variation of ecosystem indicators; Table S5: Cultivated areas and cotton yield in southern Xinjiang and the number of livestock in the Ili region; Table S6: Average correlations between ecosystem indicators and precipitation accumulated over various lengths of time in the entire Xinjiang; and Table S7: Average

correlations between ecosystem indicators and precipitation accumulated over various lengths of time for each pixel in Xinjiang.

**Author Contributions:** Y.Z.: writing—original draft, investigation, data curation, software; Y.L.: supervision; W.L.: software, writing—original draft preparation, editing; F.L.: software, performed the computations; Q.X.: writing—review and editing, conceptualization. All authors have read and agreed to the published version of the manuscript.

**Funding:** The research was supported in part by the National Natural Science Foundation of China (41875122), the Western Talents (2018XBYJRC004), the Guangdong Top Young Talents (2017TQ04Z359), the Introducing Talents to Western China Project of Chinese Academy of Sciences (Y932121), and the Natural Science Foundation of Guangdong Province, China (2021A1515011429).

**Data Availability Statement:** All data for this paper are properly cited and referred to in the reference list.

**Conflicts of Interest:** The authors declare no conflict of interest.

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
