# Peer review of "Ecological Responses to Climate Change and Human Activities in the Arid and Semi-Arid Regions of Xinjiang in China"

_remotesensing, doi:10.3390/rs14163911_

Round 1

Reviewer 1 Report

Study use BESS model dataset (GPP + ET) and MODIS LAI dataset for a huge region with Junggar basin and Tarim basin in China, on a 1km and yearly resolution for 2001-2015 to analyse response of ET, GGP and T on changing precipitation and temperature over this period. For generation of 1km pixel of P and T 66 Meteorological stations were used, also DEM and land cover data from ESA. With single data from local stations a validation for the ecosystem indicators was done. Main aims are 1) trend analysis of P+T and ecosystem indicators, 2) how P+T influence the ecosystem indicators. - So normal statistical analysis was used with trend and correlation analysis (Mann-Kendall for significance) and correlation analysis. Interesting with the the results is the residual analysis, to differentiate between explanation value of climate change (P+T) or human impact on ecosystem indicator changes.

For results and discussion more description of land cover change, more agriculture with irrigation in oasis regions, is necessary. With the aims research hypothesis is missing. - In result chapter spatial variation of ecosystem indicators and time trend (P+T) are well described and documented with figures. Only for semi-arid and sub-humid zone a clear decrease of ET (Fig. 5) exist. Astonishing  that for arid and hyper-arid zone GPP and LAI increase significantly. In discussion this must be more analysed in relation to the oasis and irrigation agriculture development! Main results with residual trends are, that CC impact is low, but human impact high on ecosystem indicators.

Discussion: rows 477-478: correlation of P with ET, GPP and LAI is less than 68% with r greater as 0,4 (see Fig.10), so conclusion is wrong. In rows 484-487 explanations for negative effect of P on ET and LAI is too simple - necessary for the subhumid and humid areas to analyse when rainfall occur (monthly) and how vegetation period is! Rows 491-494: cooling effect of vegetation - what is the meaning? Rows 497-500: changes in land cover is important - but no maps or information in the main text! (LC change with time and where?) - Of interest is to select pixels with main LUC (oasis areas?) and test them for correlation. Rows 505-507: inconsistent trend of ET and GPP and LAI - discuss this more - why? In grazing land less water available - less ET, in oasis areas more irrigation and therfore mor ET, GPP ?? So increased agricultural land is named, but not analysed specific. Rows 543-561: content belongs not direct to the analysis and results of the paper. For the reader it will be good to put one supplement table (S1, S2, S3) into the main text to explain and discuss better the role of human impact on ecosystem indictaors. In Fig. 13 explain PRE.

In total: Discuss more deeply the role of LUC (oasis areas, enlargement of irrigation, pasture degradation) on correlation P+T and ecosystem indicators. Is it different to the whole dataset for the climate zones?

Reviewer 2 Report

I had quite some comments on the first draft, and felt the authors had improved a lot in the second version, so I suggested to accept it with minor revisions. Now they kept improving according to other reviewers' comments, I felt it is good enough for publication, and no further comments.

Author Response

Thank you for your comment. I have improved the manuscript according to the opinions of other reviewers.

Reviewer 3 Report

This study focuses on evaluating how and to what extent climate variation and human activities influenced major indicators that are related to ecosystem functions and conditions in the past decades in a typical arid and semi-arid region in China.

The submission has all the important parts. However, there are some weak points that should be strengthened. Below please find specific comments/ suggestions, which waits for clarification:

- In the introduction section please, clarify what are its innovative contributions to science.

- Please include a framing map of the study area, with the location of the study area in China, and the Asian continent.

- Please include a methodological framework in the materials and methods sections.

- The methodology needs more explanations regarding alternative approaches.

- Please show the data from section 2.2 in a table. This table should include the variable, the original scale, the source, the reason for using this variable for this study, etc.

- In figure 1, coordinates and scale are missing.

- A scale is missing in figure 2.

- Please include scale and coordinates in figures 3, 4, 10, 11, and 13.

- The conclusions section should be expanded. More specifically, please expand on how this study will be addressed as a part of future research.

- Minor grammar and punctuation errors can be found throughout the text and need to be corrected.

Round 2

Reviewer 1 Report

With corrections and added text as well as new added tables paper is now improved. Here comments to the authors answers (points 1-8):

1. bare ground excluded in the analysis - was before not clear, but correlations are still low!

2. now ok, role of low temperature within the precipitation period

3. cooling effect of precipitation - ok  ? snowfall in the humid areas?

4. "Spatial distribution of statistics and ecosystem indicators does not match." Still unclear - missing explanations for this

5. now explained - ok

6. with added tables now mok

7. ok

8. added text ok

Reviewer 3 Report

Thank you and your colleagues for the modifications that you have made to this manuscript and how well you have responded to the suggestions.

This manuscript is a resubmission of an earlier submission. The following is a list of the peer review reports and author responses from that submission.

Round 1

Reviewer 1 Report

The topic treated in this article is interesting however already well debated in literature; indeed, many are the papers published until now on LAI and ET changes from satellite analysis. So, what is the real advance of this article? Unfortunately it seems just an additional GIS exercise (anyway respectable) on a study area. In addition, a detailed field validation, with photos and surveys seem not well described and clarified in the text, and this is undoubtedly a weak point. Having said that, I'm also skeptical about the definition of "ecological indicator", what the authors exactly mean about ecology? LAI? GPP? Are these enough to say ecology? what about the field validation of GPP and ecosystem services? Unfortunately because of the above critical issues, I don't think that the paper is suitable for publication.

Reviewer 2 Report

Comments on "Ecological responses to climate change and human activities in the arid and semi-arid regions of Xinjiang in China"

This manuscript tries to expand established approaches to distinguish the impacts of climate change and human activities on ecological indicators in arid regions of Xinjiang, China. The background introduction is excellent, the results, discussion and conclusion have been nicely presented, especially in the discussion section, the introduction of urbanization, ecological water transfer projects, oasis irrigation, grazing, sandstorms, and others largely help building the connection between the analysis results and the conclusions. Yet due to the following considerations, some important technical details are missing, and the foundation of this manuscript is not solid and rigorous, so I would recommend it to be accepted only after major revision.     

Major review:

1. Both precipitation and temperature data are rather important in this manuscript, yet (a) how these observation values from meteorological stations are aggregated to annual values is missing, (b) the spatial resolution after interpolation is missing, (c) how reliable are the climate data after aggregation and interpolation is missing.

2. Equations (7) and (8) are the foundation of the entire manuscript, since Eq. (7) characterizes ecological changes induced by climate change, and Eq. (8) characterizes ecological changes induced by human activities. (a) Comparing to Eq. (7) of the referenced paper Sun et al. [29], Eq. (7) of this manuscript is erroneous and over-simplified. In [29], before reaching Eq. (7), the authors introduced Eqs. (2)-(4) with detailed temporal and magnitude considerations, for example, monthly accumulation of monthly mean temperature greater than 10 degC. Yet such technical details are totally omitted in this manuscript, so the readers have no idea how the regression of Eq. (7) was performed. (b) Both referenced papers [29] and [52] mentioned that the regression model was built from climatic factors and maximum NDVI values, yet from this manuscript, no any similar mentioning of maximum ET/GPP/LAI values could be seen. (3) Further, in [52], the authors mentioned that even under the same framework, there are several approaches to calculate the maximum NDVI, and therefore the departure of the changes due to climate change and human activities, and each approach has its own advantage according to the application scenarios. Yet due to the aforementioned omissions, the manuscript compressed those possibilities into only one realization, which appears somewhat rigid, and might be misleading. 

My suggestion: explicitly mention that the calculation results from Eqs. (7) and (8) have other possibilities, according to the calculation/selection of the regressors; even it is impractical to go through each possibility in detail, it is beneficial to list some of them, and add discussions on possible different interpretation for other results.     

Essentially, the paragraph from Line 139 to Line 145, and the sentence "The relative contribution of climate change and human activities to ecological indicators were used based on the method proposed by Sun et al. [29]" are far from enough.

Minor reviews:

1. Consider removing Line 101, Lines 357-360; Consider revising the sentence in Line 160, and the sentence starting from Line 276. 

2. It is better to include a DEM map with Figure 1 to indicate the desert and Gobi region in Xinjiang, which are excluded from the research.

3. In Figure 3, 2009 could be a special year, what happened before and after that year?

4. Figure 8, what is EC data? is it eddy covariance? the abbreviation should be defined before first usage.

Reviewer 3 Report

The article is about the relationship between evapotranspiration (ET), gross primary production (GPP) and leaf area index (LAI) with air temperature and precipitation in the Xinjiang region. From these relationships, the authors identified the effect of climate change and human activities on ET, GPP and LAI. Overall, the paper has merit, but information on how it was designed is lacking. It is difficult to identify the different regions on the maps in the figures. Information is lacking on the flow towers used to validate the ET and GPP, such as their locations, how the data was processed, and whether they are in representative locations of the study area. Data from just 2 flow towers are not enough to state that the ET and GPP products are robust enough to be representative of the study area. Therefore, my recommendation is for major revision.

The Abstract is relatively well developed, but it is unclear how the authors confirmed the sentences in lines 28 to 31: “Results based on residual analysis indicated that human activities could account for over 72% variation in the changes of ecological indicators. Human activities have large impacts on vegetation change indicators in hyper-arid and arid zones and their relative contribution has a mean value of 79%. Which variable was used to describe human activities?

Practically every introduction is quite generic, highlighting some indicators of change in ecosystems and some relationships between climatic factors and vegetation dynamics. Only in the last paragraph, the authors highlight the object of study. My recommendation is for the authors to better justify why they chose to study the relationship of climatic factors and human activities in the Xinjiang region. For example, the authors could describe what human activities are carried out in the region, what the economy of the Xinjiang region represents for China, etc...

I recommend including a figure in the Study Area item.

I recommend incorporating item 3. Data and Methods into item 2. Material and Methods, as they describe the same information.

What time scale was used to estimate trends with the Mann-Kendall test? Monthly or Annual? This information is important to be in item 3.2.1.

Equation 7 makes no sense, i.e., the relationship between vegetation and precipitation is not equal to the relationship between vegetation and temperature. In addition, authors must describe the relationship between equation 8 and the representativeness of human contributions. This relationship is unclear. I recommend that you add a literature review on equations 7 and 8. I particularly find the relationship of equation 7 very strange, as temperature and precipitation change annually and ET, GPP and LAI are estimated by models.

The title of item 4.2 does not represent the content. I suggest that you indicate that it was analyzed by region of aridity.

Indicate in Figure 6 which relationships represent each map. It is difficult to recognize the regions in Figures 2, 6 and 7; therefore, I recommend that you add the outline of each region to make it easier for the reader to identify them.

Item 4.5 Evaluation of the BESS products in the Xinjiang appears out of nowhere in the article, with no mention of flux towers in Material and Methods. Some questions are left in the air: Are these flow towers representative of the analyzed regions? Where are these flow towers located? How was this Eddy covariance data handled (software used, gap filling method)?

The first paragraph of the discussion resembles advice. I recommend that you turn it off.

Reviewer 4 Report

Dear authors,

Congratulations on the manuscript that was very well constructed. I have just four  suggestions, which include a few questions:

1) The first paragraph of the discussion (lines 357 to 360) appears to be an instruction to the authors. If so, it might be interesting to delete it.

2) The authors concluded that Vegetation growth is limited by temperature or radiation in the dry sub-humid and humid zones and that precipitation and temperature interact to influence changes in ecological indicators. They also conclude that as the climate becomes more suitable for vegetative growth, the positive effects of temperature and the negative effects of precipitation increase. Without resorting to detailed climate information for the study area (in other publications or sites), the reader may ask: is the temperature range increasing, decreasing or remaining constant in the study area? Which analysis is more consistent: correlation between mean annual temperature and ecological indicators or correlation between temperature range and ecological indicators? If there is a tendency to increase or decrease this amplitude, it may be interesting to discuss this in the manuscript. Especially in relation to the effects on vegetative growth.

3) The parallelism between the lines GPPmodel = a + bGPPEC and ETmodel = a + bETEC with the lines GPPmodel = GPPEC and ETmodel = ETEC (Figure 8) suggests that the model has potential but contains a consistent error, which could perhaps be corrected. Unfortunately I don't have a suggestion on how to correct it and I can't say that there would be any change in the conclusions either. Is it possible to make some consideration about this in the discussion?

4) Are there any negative aspects in the manuscript? For example, any limitations or restrictions on conclusions?

Round 2

Reviewer 1 Report

Unfortunately I'm deeply convinced about the limits of this work and its not innovative perspective to the recent literature. Also, despite the indication of the authors in saying that field data have been provided in previous studies, sincerely I want to see here the field dataset used for validation, with some details about the human activities impact on local ecosystems. The lack of details and a substantial high-resolution field validation of the data, makes the entire work quite fragile to be presented at the scientific community. Therefore from my side, with a great respect I have with the authors, that they are very good scientists, I cannot support publication. 

Reviewer 2 Report

Comments on the updated version of "Ecological responses to climate change and human activities in 2 the arid and semi-arid regions of Xinjiang in China"

In this majorly updated version, the authors carefully addressed my former comments in a satisfactory way. They explained why they use the approach different from the cited papers, made significant modifications especially in the methodology and discussion parts, and added substantial amount of supplementary materials, which largely enriched the technical details formerly missing, and strengthened the points they wanted to raise. As a result, this article is now much more readable, and comparable to the peer literature. Together with former merits, I would recommend it to be accepted with minor revision. 

I have only one question:     

Table 1, how do you guarantee the summation of slope_pes/slope_obs + slope_res/slope_obs equals to 1? from the combination of Equations (8) and (9)? what if this summation is much smaller or greater than 1? 

Reviewer 3 Report

The authors addressed all suggestions that I had made. So, I recommend accept this paper.